# Neural system identification for large populations separating "what" and "where"

**David A. Klindt** [* 1-3], **Alexander S. Ecker** [* 1,2,4,6], **Thomas Euler** [1-3], **Matthias Bethge** [1,2,4-6]

[*] *Authors contributed equally*

[1] Centre for Integrative Neuroscience, University of Tübingen, Germany
[2] Bernstein Center for Computational Neuroscience, University of Tübingen, Germany
[3] Institute for Ophthalmic Research, University of Tübingen, Germany
[4] Institute for Theoretical Physics, University of Tübingen, Germany
[5] Max Planck Institute for Biological Cybernetics, Tübingen, Germany
[6] Center for Neuroscience and Artificial Intelligence, Baylor College of Medicine, Houston, USA

`klindt.david@gmail.com, alexander.ecker@uni-tuebingen.de,`
`thomas.euler@cin.uni-tuebingen.de, matthias.bethge@bethgelab.org`

## Abstract

Neuroscientists classify neurons into different types that perform similar computations at different locations in the visual field. Traditional methods for neural system identification do not capitalize on this separation of "what" and "where". Learning deep convolutional feature spaces that are shared among many neurons provides an exciting path forward, but the architectural design needs to account for data limitations: While new experimental techniques enable recordings from thousands of neurons, experimental time is limited so that one can sample only a small fraction of each neuron's response space. Here, we show that a major bottleneck for fitting convolutional neural networks (CNNs) to neural data is the estimation of the individual receptive field locations – a problem that has been scratched only at the surface thus far. We propose a CNN architecture with a sparse readout layer factorizing the spatial (where) and feature (what) dimensions. Our network scales well to thousands of neurons and short recordings and can be trained end-to-end. We evaluate this architecture on ground-truth data to explore the challenges and limitations of CNN-based system identification. Moreover, we show that our network model outperforms current state-of-the art system identification models of mouse primary visual cortex.

## 1 Introduction

In neural system identification, we seek to construct quantitative models that describe how a neuron responds to arbitrary stimuli [1, 2]. In sensory neuroscience, the standard way to approach this problem is with a generalized linear model (GLM): a linear filter followed by a point-wise nonlinearity [3, 4]. However, neurons elicit complex nonlinear responses to natural stimuli even as early as in the retina [5, 6] and the degree of nonlinearity increases as ones goes up the visual hierarchy. At the same time, neurons in the same brain area tend to perform similar computations at different positions in the visual field. This separability of what is computed from where it is computed is a key idea underlying the notion of functional cell types tiling the visual field in a retinotopic fashion.

For early visual processing stages like the retina or primary visual cortex, several nonlinear methods have been proposed, including energy models [7, 8], spike-triggered covariance methods [9, 10], linear-nonlinear (LN-LN) cascades [11, 12], convolutional subunit models [13, 14] and GLMs based on handcrafted nonlinear feature spaces [15]. While these models outperform the simple GLM, they

still cannot fully account for the responses of even early visual processing stages (i.e. retina, V1), let alone higher-level areas such as V4 or IT. The main problem is that the expressiveness of the model (i.e. number of parameters) is limited by the amount of data that can be collected for each neuron.

The recent success of deep learning in computer vision and other fields has sparked interest in using deep learning methods for understanding neural computations in the brain [16, 17, 18], including promising first attempts to learn feature spaces for neural system identification [19, 20, 21, 22, 23]. In this study, we would like to achieve a better understanding of the possible advantages of deep learning methods over classical tools for system identification by analyzing their effectiveness on ground truth models. Classical approaches have traditionally been framed as individual multivariate regression problems for each recorded neuron, without exploiting computational similarities between different neurons for regularization. One of the most obvious similarities between different neurons, however, is that the visual system simultaneously extracts similar features at many different locations. Because of this spatial equivariance, the same nonlinear subspace is spanned at many nearby locations and many neurons share similar nonlinear computations. Thus, we should be able to learn much more complex nonlinear functions by combining data from many neurons and learning a common feature space from which we can linearly predict the activity of each neuron.

We propose a convolutional neural network (CNN) architecture with a special readout layer that separates the problem of learning a common feature space from estimating each neuron's receptive field location and cell type, but can still be trained end-to-end on experimental data. We evaluate this model architecture using simple simulations and show its potential for developing a functional characterization of cell types. Moreover, we show that our model outperforms the current state-of-the-art on a publicly available dataset of mouse V1 responses to natural images [19].

## 2 Related work

Using artificial neural networks to predict neural responses has a long history [24, 25, 26]. Recently, two studies [13, 14] fit two-layer models with a convolutional layer and a pooling layer. They do find marked improvements over GLMs and spike-triggered covariance methods, but like most other previous studies they fit their model only to individual cells' responses and do not exploit computational similarities among neurons.

Antolik et al. [19] proposed learning a common feature space to improve neural system identification. They outperform GLM-based approaches by fitting a multi-layer neural network consisting of parameterized difference-of-Gaussian filters in the first layer, followed by two fully-connected layers. However, because they do not use a convolutional architecture, features are shared only locally. Thus, every hidden unit has to be learned 'from scratch' at each spatial location and the number of parameters in the fully-connected layers grows quadratically with population size.

McIntosh et al. [20] fit a CNN to retinal data. The bottleneck in their approach is the final fully-connected layer that maps the convolutional feature space to individual cells' responses. The number of parameters in this final readout layer grows very quickly and even for their small populations represents more than half of the total number of parameters.

Batty et al. [21] also advocate feature sharing and explore using recurrent neural networks to model the shared feature space. They use a two-step procedure, where they first estimate each neuron's location via spike-triggered average, then crop the stimulus accordingly for each neuron and then learn a model with shared features. The performance of this approach depends critically on the accuracy of the initial location estimate, which can be problematic for nonlinear neurons with a weak spike-triggered average response (e. g. complex cells in primary visual cortex).

Our contribution is a novel network architecture consisting of a number of convolutional layers followed by a sparse readout layer factorizing the spatial and feature dimensions. Our approach has two main advantages over prior art. First, it reduces the effective number of parameters in the readout layer substantially while still being trainable end-to-end. Second, our readout forces all computations to be performed in the convolutional layers while the factorized readout layer provides an estimate of the receptive field location and the cell type of each neuron.

In addition, our work goes beyond the findings of these previous studies by providing a systematic evaluation, on ground truth models, of the advantages of feature sharing in neural system identification – in particular in settings with many neurons and few observations.

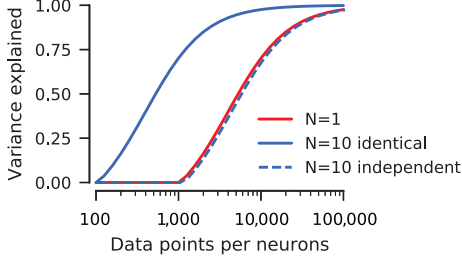

Figure 1: Feature sharing makes more efficient use of the available data. Red line: System identification performance with one recorded neuron. Blue lines: Performance for a hypothetical population of 10 neurons with identical receptive field shapes whose locations we know. A shared model (solid blue) is equivalent to having $10\times$ as much data, i. e. the performance curve shifts to the left. If we fit all neurons independently (dashed blue), we do not benefit from their similarity.

## 3  Learning a common feature space

We illustrate why learning a common feature space makes much more efficient use of the available data by considering a simple thought experiment. Suppose we record from ten neurons that all compute exactly the same function, except that they are located at different positions. If we know each neuron's position, we can pool their data to estimate a single model by shifting the stimulus such that it is centered on each neuron's receptive field. In this case we have effectively ten times as much data as in the single-neuron case (Fig. 1, red line) and we will achieve the same model performance with a tenth of the data (Fig. 1, solid blue line). In contrast, if we treat each neuron as an individual regression problem, the performance will on average be identical to the single-neuron case (Fig. 1, dashed blue line). Although this insight has been well known from transfer learning in machine learning, it has so far not been applied widely in a neuroscience context.

In practice we neither know the receptive field locations of all neurons a priori nor do all neurons implement exactly the same nonlinear function. However, the improvements of learning a shared feature space can still be substantial. First, estimating the receptive field location of an individual neuron is a much simpler task than estimating its entire nonlinear function from scratch. Second, we expect the functional response diversity within a cell type to be much smaller than the overall response diversity across cell types [27, 28]. Third, cells in later processing stages (e. g. V1) share the nonlinear computations of their upstream areas (retina, LGN), suggesting that equipping them with a common feature space will simplify learning their individual characteristics [19].

## 4  Feature sharing in a simple linear ground-truth model

We start by investigating the possible advantages of learning a common feature space with a simple ground truth model – a population of linear neurons with Poisson-like output noise:

$$r_n = \mathbf{a}_n^{\mathrm{T}}\mathbf{s} \qquad y_n \sim \mathcal{N}\left(r_n, \sqrt{|r_n|}\right) \tag{1}$$

Here, $s$ is the (Gaussian white noise) stimulus, $r_n$ the firing rate of neuron $n$, $a_n$ its receptive field kernel and $y_n$ its noisy response. In this simple model, the classical GLM-based approch reduces to (regularized) multivariate linear regression, which we compare to a convolutional neural network.

### 4.1  Convolutional neural network model

Our neural network consists of a convolutional layer and a readout layer (Fig. 2). The first layer convolves the image with a number of kernels to produce $K$ feature maps, followed by batch normalization [29]. There is no nonlinearity in the network (i.e. activation function is the identity). Batch normalization ensures that the output has fixed variance, which is important for the regularization in the second layer. The readout layer pools the output, $c$, of the convolutional layer by applying a sparse mask, $q$, for each neuron:

$$\hat{r}_n = \sum_{i,j,k} c_{ijk} q_{ijkn} \tag{2}$$

Here, $\hat{r}_n$ is the predicted firing rate of neuron $n$. The mask $q$ is factorized in the spatial and feature dimension:

$$q_{ijkn} = m_{ijn} w_{kn}, \tag{3}$$

where $m$ is a spatial mask and $w$ is a set of $K$ feature weights for each neuron. The spatial mask and feature weights encode each neuron's receptive field location and cell type, respectively. As we expect them to be highly sparse, we regularize both by an L1 penalty (with strengths $\lambda_m$ and $\lambda_w$).

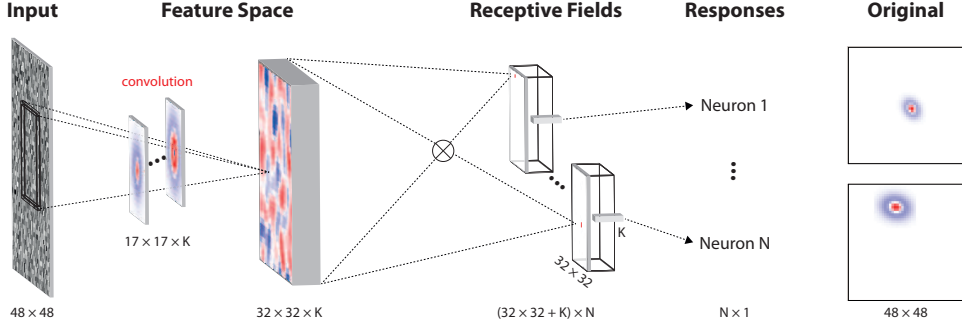

Figure 2: Our proposed CNN architecture in its simplest form. It consists of a feature space module and a readout layer. The feature space is extracted via one or more convolutional layers (here one is shown). The readout layer computes for each neuron a weighted sum over the entire feature space. To keep the number of parameters tractable and facilitate interpretability, we factorize the readout into a location mask and a vector of feature weights, which are both encouraged to be sparse by regularizing with L1 penalty.

By factorizing the spatial and feature dimension in the readout layer, we achieve several useful properties: first, it reduces the number of parameters substantially compared to a fully-connected layer [20]; second, it limits the expressiveness of the layer, forcing the 'computations' down to the convolutional layers, while the readout layer performs only the selection; third, this separation of computation from selection facilitates the interpretation of the learned parameters in terms of functional cell types.

We minimize the following penalized mean-squared error using the Adam optimizer [30]:

$$\mathcal{L} = \frac{1}{B} \sum_{b,n} (y_{bn} - \hat{r}_{bn})^2 + \lambda_m \sum_{i,j,n} |m_{ijn}| + \lambda_w \sum_{k,n} |w_{kn}| \qquad (4)$$

where $b$ denotes the sample index and $B = 256$ is the minibatch size. We use an initial learning rate of 0.001 and early stopping based on a separate validation set consisting of 20% of the training set. When the validation error has not improved for 300 consecutive steps, we go back to the best parameter set and decrease the learning rate once by a factor of ten. After the second time we end the training. We find the optimal regularization weights $\lambda_m$ and $\lambda_w$ via grid search.

To achieve optimal performance, we found it to be useful to initialize the masks well. Shifting the convolution kernel by one pixel in one direction while shifting the mask in the opposite direction in principle produces the same output. However, because in practice the filter size is finite, poorly initialized masks can lead to suboptimal solutions with partially cropped filters (cf. Fig. 3C, $CNN_{10}$). To initialize the masks, we calculated the spike-triggered average for each neuron, smoothed it with a large Gaussian kernel and took the pixel with the maximum absolute value as our initial guess for the neurons' location. We set this pixel to the standard deviation of the neuron's response (because the output of the convolutional layer has unit variance) and initialized the rest of the mask randomly from a Gaussian $\mathcal{N}(0, 0.001)$. We initialized the convolution kernels randomly from $\mathcal{N}(0, 0.01)$ and the feature weights from $\mathcal{N}(1/K, 0.01)$.

## 4.2 Baseline models

In the linear example studied here, the GLM reduces to simple linear regression. We used two forms of regularization: lasso (L1) and ridge (L2). To maximize the performance of these baseline models, we cropped the stimulus around each neuron's receptive field. Thus, the number of parameters these models have to learn is identical to those in the convolution kernel of the CNN. Again, we cross-validated over the regularization strength.

## 4.3 Performance evaluation

To measure the models' performance we compute the fraction of explainable variance explained:

$$\text{FEV} = 1 - \left\langle (\hat{r} - r)^2 \right\rangle / \text{Var}(r) \qquad (5)$$

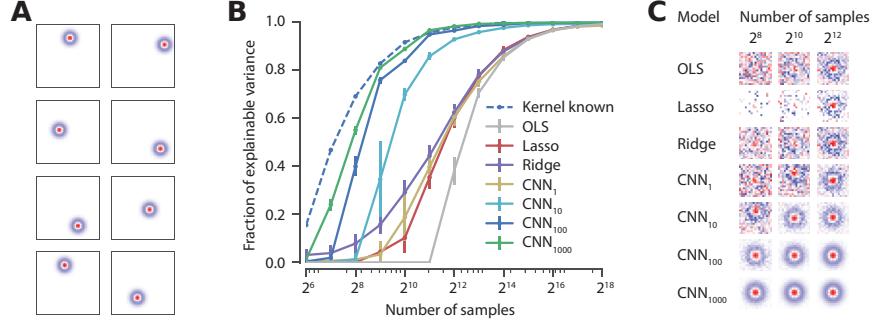

Figure 3: Feature sharing in homogeneous linear population. **A**, Population of homogeneous spatially shifted on-center/off-surround neurons. **B**, Model comparison: Fraction of explainable variance explained vs. the number of samples used for fitting the models. Ordinary least squares (OLS), L1 (Lasso) and L2 (Ridge) regularized regression models are fit to individual neurons. $\text{CNN}_N$ are convolutional models with $N$ neurons fit jointly. The dashed line shows the performance (for $N \to \infty$) of estimating the mask given the ground truth convolution kernel.**C**, Learned filters for different methods and number of samples.

which is evaluated on the ground-truth firing rates $r$ without observation noise. A perfect model would achieve FEV = 1. We evaluate FEV on a held-out test set not seen during model fitting and cross-validation.

## 4.4 Single cell type, homogeneous population

We first considered the idealized situation where all neurons share the same $17 \times 17$ px on-center/off-surround filter, but at different locations (Fig. 3A). In other words, there is only one feature map in the convolutional layer ($K = 1$). We used a $48 \times 48$ px Gaussian white noise stimulus and scaled the neurons' output such that $\langle |r| \rangle = 0.1$, mimicking a neurally-plausible signal-to-noise ratio at firing rates of 1 spike/s and an observation window of 100 ms. We simulated populations of $N = 1$, 10, 100 and 1000 neurons and varied the amount of training data.

The CNN model consistently outperformed the linear regression models (Fig. 3B). The ridge-regularized linear regression explained around 60% of the explainable variance with 4,000 samples (i.e. pairs of stimulus and $N$-dimensional neural response vector). A CNN model pooling over 10 neurons achieved the same level of performance with less than a quarter of the data. The margin in performance increased with the number of neurons pooled over in the model, although the relative improvement started to level off when going from 100 to 1,000 neurons.

With few observations, the bottleneck appears to be estimating each neuron's location mask. Two observations support this hypothesis. First, the $\text{CNN}_{1000}$ model learned much 'cleaner' weights with 256 samples than ridge regression with 4,096 (Fig. 3C), although the latter achieved a higher predictive performance (FEV = 55% vs. 65%). This observation suggests that the feature space can be learned efficiently with few samples and many neurons, but that the performance is limited by the estimation of neurons' location masks. Second, when using the ground-truth kernel and optimizing solely the location masks, performance was only marginally better than for 1,000 neurons (Fig. 3B, blue dotted line), indicating an upper performance bound by the problem of estimating the location masks.

## 4.5 Functional classification of cell types

Our next step was to investigate whether our model architecture can learn interpretable features and obtain a functional classification of cell types. Using the same simple linear model as above, we simulated two cell types with different filter kernels. To make the simulation a bit more realistic, we made the kernels heterogeneous within a cell type (Fig. 4A). We simulated a population of 1,000 neurons (500 of each type).

With sparsity on the readout weights every neuron has to select one of the two convolutional kernels. As a consequence, the feature weights represent more or less directly the cell type identity of each

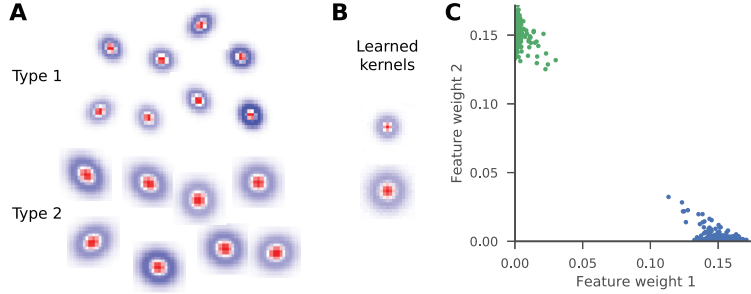

Figure 4: **A**, Example receptive fields of two types of neurons, differing in their average size. **B**, Learned filters of the CNN model. **C**, Scatter plot of the feature weights for the two cell types.

neuron (Fig. 4C). This in turn forces the kernels to learn the average of each type (Fig. 4B). However, any other set of kernels spanning the same subspace would have achieved the same predictive performance. Thus, we find that sparsity on the feature weights facilitates interpretability: each neuron chooses one feature channel which represents the essential computation of this type of neuron.

## 5 Learning nonlinear feature spaces

### 5.1 Ground truth model

Next, we investigated how our approach scales to more complex, nonlinear neurons and natural stimuli. To keep the benefits of having ground truth data available, we chose our model neurons from the VGG-19 network [31], a popular CNN trained on large-scale object recognition. We selected four random feature maps from layer conv2_2 as 'cell types'. For each cell type, we picked 250 units with random locations ($32 \times 32$ possible locations). We computed ground-truth responses for all 1000 cells on $44 \times 44$ px image patches obtained by randomly cropping images from the ImageNet (ILSVRC2012) dataset. As before, we rescaled the output to produce sparse, neurally plausible mean responses of 0.1 and added Poisson-like noise.

We fit a CNN with three convolutional layers consisting of 32, 64 and 4 feature maps (kernel size $5 \times 5$), followed by our sparse, factorized readout layer (Fig. 5A). Each convolutional layer was followed by batch normalization and a ReLU nonlinearity. We trained the model using Adam with a batch-size of 64 and the same initial step size, early stopping, cross-validation and initialization of the masks as described above. As a baseline, we fit a ridge-regularized GLM with ReLU nonlinearity followed by an additional bias.

To show that our sparse, factorized readout layer is an important feature of our architecture, we also implemented two alternative ways of choosing the readout, which have been proposed in previous work on learning common feature spaces for neural populations. The first approach is to estimate the receptive field location in advance based on the spike-triggered average of each neuron [21].[1] To do so, we determined the pixel with the strongest spike-triggered average. We then set this pixel to one in the location mask and all other pixels to zero. We then kept the location mask fixed while optimizing convolution kernels and feature weights. The second approach is to use a fully-connected readout tensor [20] and regularize the activations of all neurons with L1 penalty. In addition, we regularized the fully-connected readout tensor with L2 weight decay. We fit both models to populations of 1,000 neurons.

Our CNN with the factorized readout outperformed all three baselines (Fig. 5B).[2] The performance of the GLM saturated at $\approx$20% FEV (Fig. 5B), highlighting the high degree of nonlinearity of our model neurons. Using a fully-connected readout [20] incurred a substantial performance penalty when the number of samples was small and only asymptotically (for a large number of samples) reached the same performance as our factorized readout. Estimating the receptive field location in

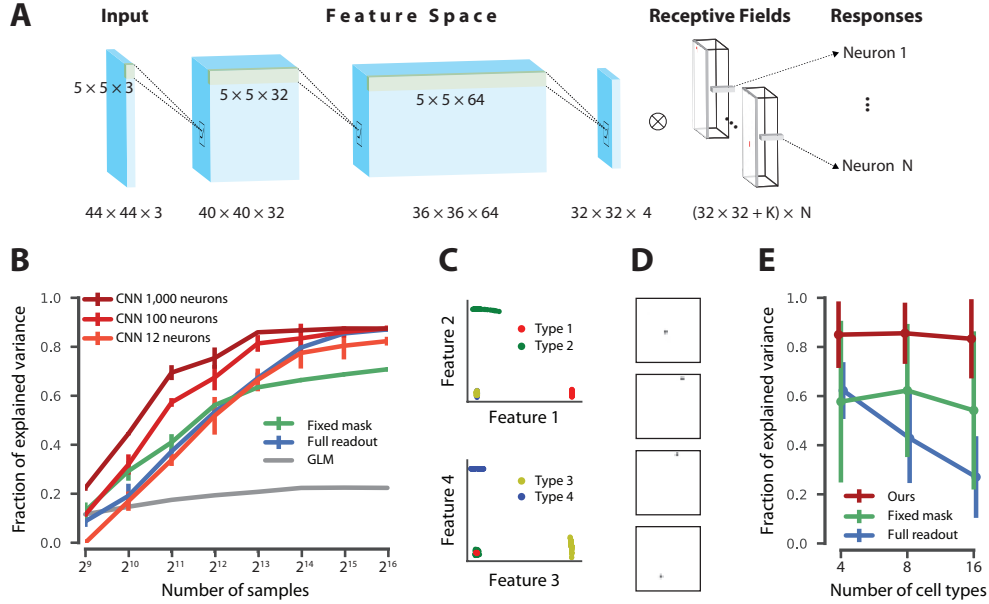

Figure 5: Inferring a complex, nonlinear feature space. **A**, Model architecture. **B**, Dependence of model performance (FEV) on number of samples used for training. **C**, Feature weights of the four cell types for $\text{CNN}_{1000}$ with $2^{15}$ samples cluster strongly. **D**, Learned location masks for four randomly chosen cells (one per type). **E**, Dependence of model performance (FEV) on number of types of neurons in population, number of samples fixed to $2^{12}$.

advance [21] led to a drop in performance – even for large sample sizes. A likely explanation for this finding is the fact that the responses are quite nonlinear and, thus, estimates of the receptive field location via spike-triggered average (a linear method) are not very reliable, even for large sample sizes.

Note that the fact that we can fit the model is not trivial, although ground truth is a CNN. We have observations of noise-perturbed VGG units whose locations we do not know. Thus, we have to infer both the location of each unit as well as the complex, nonlinear feature space simultaneously. Our results show that our model solves this task more efficiently than both simpler (GLM) and equally expressive [20] models when the number of samples is relatively small.

In addition to fitting the data well, the model also recovered both the cell types and the receptive field locations correctly (Fig. 5C, D). When fit using $2^{16}$ samples ($2^{10}$ for validation/test and the rest for training), the readout weights of the four cell types clustered nicely (Fig. 5C) and it successfully recovered the location masks (Fig. 5D). In fact, all cells were classified correctly based on their largest feature weight.

Next, we investigated how our model and its competitors [20, 21] fare when scaling up to large recordings with many types of neurons. To simulate this scenario, we sampled again VGG units (from the same layer as above), taking 64 units with random locations from up to 16 different feature maps (i.e. cell types). Correspondingly we increased the number of feature maps in the last convolutional layer of the models. We fixed the number of training samples to $2^{12}$ to compare models in a challenging regime (cf. Fig. 5B) where performance can be high but is not yet asymptotic.

Our CNN model scales gracefully to more diverse neural populations (Fig. 5E), remaining roughly at the same level of performance. Similarly, the CNN with the fixed location masks estimated in advance scales well, although with lower overall performance. In contrast, the performance of the fully-connected readout drops fast, because the number of parameters in the readout layer grows very quickly with the number of feature maps in the final convolutional layer. In fact, we were unable to fit models with more than 16 feature maps with this approach, because the size of the read-out tensor became prohibitively large for GPU memory.

Table 1: Application to data from primary visual cortex (V1) of mice [19]. The table shows average correlations between model predictions and neural responses on the test set.

| Scan | 1 | 2 | 3 | Average |
|---|---|---|---|---|
| Antolik et al. 2016 [19] | 0.51 | 0.43 | 0.46 | 0.47 |
| LNP | 0.37 | 0.30 | 0.38 | 0.36 |
| CNN with fully connected readout | 0.47 | 0.34 | 0.43 | 0.43 |
| CNN with fixed mask | 0.45 | 0.38 | 0.41 | 0.42 |
| CNN with factorized readout (ours) | **0.55** | **0.45** | **0.49** | **0.50** |

Finally, we asked how far we can push our model with long recordings and many neurons. We tested our model with $2^{16}$ training samples from 128 different types of neurons (again 64 units each). On this large dataset with $\approx 60.000$ recordings from $\approx 8.000$ neurons we were still able to fit the model on a single GPU and perform at 90% FEV (data not shown). Thus, we conclude that our model scales well to large-scale problems with thousands of nonlinear and diverse neurons.

## 5.2 Application to data from primary visual cortex

To test our approach on real data and going beyond the previously explored retinal data [20, 21], we used the publicly available dataset from Antolik et al. [19].[3] The dataset has been obtained by two-photon imaging in the primary visual cortex of sedated mice viewing natural images. It contains three scans with 103, 55 and 102 neurons, respectively, and their responses to static natural images. Each scan consists of a training set of images that were each presented once (1800, 1260 and 1800 images, respectively) as well as a test set consisting of 50 images (each image repeated 10, 8 and 12 times, respectively). We use the data in the same form as the original study [19], to which we refer the reader for full details on data acquisition, post-processing and the visual stimulation paradigm.

To fit this dataset, we used the same basic CNN architecture described above, with three small modifications. First, we replaced the ReLU activation functions by a soft-thresholding nonlinearity, $f(x) = \log(1 + \exp(x))$. Second, we replaced the mean-squared error loss by a Poisson loss (because neural responses are non-negative and the observation noise scales with the mean response). Third, we had to regularize the convolutional kernels, because the dataset is relatively limited in terms of recording length and number of neurons. We used two forms of regularization: smoothness and group sparsity. Smoothness is achieved by an L2 penalty on the Laplacian of the convolution kernels:

$$\mathcal{L}_{\text{laplace}} = \lambda_{\text{laplace}} \sum_{i,j,k,l} (W_{:,:,kl} * L)_{ij}^2, \qquad L = \begin{bmatrix} 0.5 & 1 & 0.5 \\ 1 & -6 & 1 \\ 0.5 & 1 & 0.5 \end{bmatrix} \qquad (6)$$

where $W_{ijkl}$ is the 4D tensor representing the convolution kernels, $i$ and $j$ depict the two spatial dimensions of the filters and $k, l$ the input and output channels. Group sparsity encourages filters to pool from only a small set of feature maps in the previous layer and is defined as:

$$\mathcal{L}_{\text{group}} = \lambda_{\text{group}} \sum_{i,j} \sqrt{\sum_{kl} W_{ijkl}^2}. \qquad (7)$$

We fit CNNs with one, two and three layers. After an initial exploration of different CNN architectures (filter sizes, number of feature maps) on the first scan, we systematically cross-validated over different filter sizes, number of feature maps and regularization strengths via grid search on all three scans. We fit all models using 80% of the training dataset for training and the remaining 20% for validation using Adam and early stopping as described above. For each scan, we selected the best model based on the likelihood on the validation set. In all three scans, the best model had 48 feature maps per layer and $13 \times 13$ px kernels in the first layer. The best model for the first two scans had $3 \times 3$ kernels in the subsequent layers, while for the third scan larger $8 \times 8$ kernels performed best.

We compared our model to four baselines: (a) the Hierarchical Structural Model from the original paper publishing the dataset [19], (b) a regularized linear-nonlinear Poisson (LNP) model, (c) a CNN with fully-connected readout (as in [20]) and (d) a CNN with fixed spatial masks, inferred

from the spike-triggered averages of each neuron (as in [21]). We used a separate, held-out test set to compare the performance of the models. On the test set, we computed the correlation coefficient between the response predicted by each model and the average observed response across repeats of the same image.[4]

Our CNN with factorized readout outperformed all four baselines on all three scans (Table 1). The other two CNNs, which either did not use a factorized readout (as in [20]) or did not jointly optimize feature space and readout (as in [21]), performed substantially worse. Interestingly, they did not even reach the performance of [19], which uses a three-layer fully-connected neural network instead of a CNN. Thus, our model is the new state of the art for predicting neural responses in mouse V1 and the factorized readout was necessary to outperform an earlier (and simpler) neural network architecture that also learned a shared feature space for all neurons [19].

## 6  Discussion

Our results show that the benefits of learning a shared convolutional feature space can be substantial. Predictive performance increases, however, only until an upper bound imposed by the difficulty of estimating each neuron's location in the visual field. We propose a CNN architecture with a sparse, factorized readout layer that separates these two problems effectively. It allows scaling up the complexity of the convolutional layers to many parallel channels (which are needed to describe diverse, nonlinear neural populations), while keeping the inference problem of each neuron's receptive field location and type identity tractable.

Furthermore, our performance curves (see Figs. 3 and 5) may inform experimental designs by determining whether one should aim for longer recordings or more neurons. For instance, if we want to explain at least 80% of the variance in a very homogenous population of neurons, we could choose to record either $\approx 2,000$ responses from 10 cells or $\approx 500$ responses from 1,000 cells.

Besides making more efficient use of the data to infer their nonlinear computations, the main promise of our new regularization scheme for system identification with CNNs is that the explicit separation of "what" and "where" provides us with a principled way to functionally classify cells into different types: the feature weights of our model can be thought of as a "barcode" identifying each cell type. We are currently working on applying this approach to large-scale data from the retina and primary visual cortex. Later processing stages, such as primary visual cortex could additionally benefit from similarly exploiting equivariance not only in the spatial domain, but also (approximately) in the orientation or direction-of-motion domain.

**Availability of code**

The code to fit the models and reproduce the figures is available online at:
`https://github.com/david-klindt/NIPS2017`

**Acknowledgements**

We thank Philipp Berens, Katrin Franke, Leon Gatys, Andreas Tolias, Fabian Sinz, Edgar Walker and Christian Behrens for comments and discussions.

This work was supported by the German Research Foundation (DFG) through Collaborative Research Center (CRC 1233) "Robust Vision" as well as DFG grant EC 479/1-1; the European Union's Horizon 2020 research and innovation programme under the Marie Skłodowska-Curie grant agreement No 674901; the German Excellency Initiative through the Centre for Integrative Neuroscience Tübingen (EXC307). The research was also supported by Intelligence Advanced Research Projects Activity (IARPA) via Department of Interior/Interior Business Center (DoI/IBC) contract number D16PC00003. The U.S. Government is authorized to reproduce and distribute reprints for Governmental purposes notwithstanding any copyright annotation thereon. Disclaimer: The views and conclusions contained herein are those of the authors and should not be interpreted as necessarily representing the official policies or endorsements, either expressed or implied, of IARPA, DoI/IBC, or the U.S. Government.

## Footnotes

[1]Note that they used a recurrent neural network for the shared feature space. Here we only reproduce their approach to defining the readout.

[2]It did not reach 100% performance, since the feature space we fit was smaller and the network shallower than the one used to generate the ground truth data.

[3]See [22, 23] for concurrent work on primate V1.

[4]We used the correlation coefficient for evaluation (a) to facilitate comparison with the original study [19] and (b) because estimating FEV on data with a small number of repetitions per image is unreliable.

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
