[Reviews · NeurIPS 2017]

Reviewer 1



Summary: This paper presents a CNN architecture suitable for fitting neural data from visual cortices where the same features are repeated at many locations in the visual field. The method replaces the standard fully connected layer at the end of deep CNNs with a sparse factorized layer that explains each neuron’s response in terms of a location and a ‘cell type’. A progression of more complex experiments on simulated data show that the system can correctly recover neural RFs with fewer samples. Finally, the approach is applied to calcium imaging data from mouse primary visual cortex, where it improves on the state of the art by 10%. Major comments: This paper presents a promising approach to neural system identification. The main architectural innovation, the factorization of the densely connected layer into sparse what and where components, is well-motivated neurally and could see widespread use in the future. The experiments are done to an extremely high standard, and convincingly show that the proposed approach can recover correct response mappings on challenging datasets. The visual cortex results, for instance, contain a true test set which the model will be evaluated on only after review. This is a model of rigorous methodology. The paper also reports a variety of small tricks that may also be valuable to practitioners, such as their weight initialization scheme based on spike triggered averages.

Reviewer 2



The main idea describes in this paper is to exploit a key property of the organization of neurons in our visual cortex to improve the fitting of CCNs to neural data. The main idea is to apply a mask over space and features on the output of feature maps is a good one.o This allows to exploit the convolutional component of the architecture and lower the sample complexity of the fitting procedure. In general, the experiments are carried out well and the paper is technically sound. My main criticism of this paper is that the overall contribution of the paper remains relatively limited. It is likely that the approach will constitute an important technical contribution as part of a larger study but by itself, the significance of the approach remains too limited to justify publication. Minor point: spike-triggered covariance is a method, not a model (see pp. 1)

Reviewer 3



I am no expert in this field, so right now I can not fully judge on the potential impact of the paper at hand in this field of science. So first I want to ask a few clarifying questions to the authors to give me a clearer picture. I will also give a preliminary judgement at the end. Introuction 34-47: I think it should be mentioned that using more expressive models such like CNNs will come at a cost. Simple models such as GLMs are very interpretable, it also easier to argue for biological plausibility than with neural network trained by backpropagation. I do agree with your point about:"Thus, we should be able to learn much more complex nonlinear functions by combining data from many neurons and learning a common feature space from which we can linearly predict the activity of each neuron." Related Work: The approaches of McIntosh et al.[17] and Batty et al. [18] seem to be the most similar to yours. What were the practical and theoretical reasons for not comparing to them in your experiments? 3 Learning a common feature space This section describes a well-known insight from transfer learning in ML. Maybe establishing a link here is useful as you use a ML technique for a neuroscience application. 5 Learning non-linear feature spaces Is this comparison to GLMs fair? Your proposed architecture seems to share a lot of properties with the ground truth (such as convolutional layers). 5.2 Application to data from primary visual cortex It seems you needed to make a few modifications to your architecture. Can you respond to a possible çritique that says your approach only works better because you have more degrees of freedom to adjust hyperparameters to a given task? Without these adjustments, were you still able to improve over previous state of the art? -- The paper is well written and my initial impression is that this is a good contribution. The intuition to combining data from many neurons and learning a common feature space is intuitive and appears straightforward. I can't fully judge if this is novel or incremental, compared to the works of McIntosh et al.[17] and Batty et al. and I have some open questions with regards to the experiments.